# Reasons for Ineffectiveness in Improving Dewaterability of Anaerobically Digested Sludge by Bioleaching

**Haochi Zhang** [1,†], **Dejin Zhang** [1,†], **Yujun Zhou** [2], **Di Fang** [1], **Chunhong Cui** [1], **Jianru Liang** [1], **Bo Zhou** [1], **Mingjiang Zhang** [1], **Jiansheng Li** [2] and **Lixiang Zhou** [1,3,*]

1   Department of Environmental Engineering, College of Resources and Environmental Sciences, Nanjing Agricultural University, Nanjing 210095, China
2   Jiangsu Key Laboratory of Chemical Pollution Control and Resources Reuse, School of Environmental and Biological Engineering, Nanjing University of Science and Technology, Nanjing 210094, China
3   Jiangsu Collaborative Innovation Center for Solid Organic Waste Resource Utilization, Nanjing 210095, China
*   Correspondence: lxzhou@njau.edu.cn; Tel.: +86-25-84395160; Fax: +86-25-84395160
†   These authors contributed equally to this work.

**Abstract:** The use of bioleaching for anaerobically digested sludge (ADS) was found to be ineffective compared to using it for undigested sludge (UDS) for reasons elucidated in this study. Results showed that specific resistance to filtration of ADS increased during bioleaching. The pH value of ADS increased to 7.97 and remained unchanged during bioleaching, while it decreased to 2.98 for UDS. Added $Fe^{2+}$ was not detected as the energy source for ADS. Higher alkalinity and unavailable $Fe^{2+}$ in ADS prevented the growth of the *Acidithiobacillus* species. It was found that sludge pH increased to 8.40 and then stayed within an alkaline range, whereas slime EPS content rapidly increased to 8.13 mg DOC/g VSS. These results indicated that aeration seriously deteriorated the dewaterability of ADS through bioleaching due to the unexpected drastic increase of sludge pH and slime EPS content.

**Keywords:** bioleaching; anaerobically digested sludge; undigested sludge; dewaterability; extracellular polymeric substances





## 1. Introduction

Conventional activated sludge processes have been widely used in the treatment of municipal sewage by most wastewater treatment plants (WWTPs), accompanied by the production of waste-activated sludge. For obtaining biomass energy and stabilizing concentrated waste-activated sludge (hereafter referred to as undigested sludge, UDS), anaerobic digestion processes have been widely used by WWTPs in many countries around the world [1], which has eventually led to the generation of anaerobically digested sludge (ADS). At present, less than 5% of municipal WWTPs in China have effectively operated anaerobic digestion facilities [2]. Undoubtedly, more WWTPs of sludge anaerobic digestion facilities will be constructed and operated in the near future for recovery of biogas, which will be accompanied by the generation of more ADS. Sludge dewatering in WWTPs is an essential step for subsequent disposal and reutilization of sludge by composting, landfill, incineration, etc. [3,4]. The moisture content of dewatered sludge remains up to 80% through adding polyacrylamide flocculants followed by centrifugation or filter pressing [5]. Dehydrating sludge to below 60% of moisture content will drastically reduce the amount to be disposed of and is beneficial for subsequent sludge disposal due to lower moisture content and increased calorific value [6].

The bioleaching process using the *Acidithiobacillus* species, previously explored for removing sludge-borne heavy metals [7–9], is now being applied to improving dewaterability of sludge as a novel bio-conditioning technique [10–12]. In this process, bacteria from the genus *Acidithiobacillus* (such as *Acidithiobacillus ferrooxidans*) are capable of oxidizing ferrous iron to ferric iron and lowering the sludge pH, thus creating favorable conditions

for flocculation and dewatering of the sludge flocs. Previous studies have shown that the bioleaching process followed by a diaphragm pressure filter can reduce the moisture content of undigested waste-activated sludge to less than 60% without adding macromolecular flocculants, e.g., polyacrylamide. More than 20 sludge bioleaching plants, with a total processing capacity of nearly 6000 t/d of sludge (equivalent to 80% moisture content), have realized commercial operations in China since 2010 [10,12]. The moisture content of bioleached sludge after mechanical dewatering was easily decreased to 60% or below [13]. It has been well documented that the sludge bioleaching process mainly depended on the microbial activity of the *Acidithiobacillus* species at a pH in the range of 2.0–5.0. It is widely accepted that, for the improvement of sludge dewaterability, a slightly acidic condition (e.g., pH 4.0–5.0) is usually required in the bioleaching process [10], which is obviously different from the extremely acidic condition (e.g., pH 2.0–3.5) required for the considerable removal of heavy metals from sludge [14]. Since ferrous iron added as energy substances can be oxidized and hydrolyzed to produce protons ($H^+$) during the bioleaching process, the surface negatively-charged sludge particles are neutralized [15]. Thus, the surface uncharged sludge particles will not repel each other, which was conducive to sludge dewatering. Furthermore, the extracellular polymeric substances (EPS) of sludge were thought to be responsible for the poor dewatering performance of UDS due to the presence of spatial force and interstitial water within sludge flocs [16]. The more hydrophilic EPS is produced, the more difficult it is to dehydrate the sludge [10]. Notably, EPS can be drastically reduced during bioleaching, which can enhance the dewaterability of UDS [17,18]. During the bioleaching process with the inoculation of two *Acidithiobacillus* species and the addition of $Fe^{2+}$ and $S^0$ as the energy substances, Huo et al. found that sludge slime EPS content and CST drastically decreased from 7.32 mg/g VSS and 20.50 s to 2.42 mg/g VSS and 13.70 s in the first 24 h, respectively [17]. After that, sludge slime EPS content and CST steadily increased to 18.14 mg/g VSS and 24.10 s after 96 h of bioleaching, demonstrating that sludge dewaterability was negatively correlated with the slime EPS content.

In bioleaching with the co-inoculation of *Acidithiobacillus thiooxidans* TS6 and *Acidithiobacillus ferrooxidans* LX5 and supplement of 2 g/L $S^0$ and 2.4 g/L $Fe^{2+}$, the dewaterability of UDS could be improved by decreasing SRF and CST by 93.1% and 74.1% after 42 h of bioleaching, respectively [13]. However, bioleaching for ADS required a much longer time to acclimate bacteria. For example, at total solid content as low as 1% and bioleaching time as long as 18 days, the dewaterability of ADS could be improved, as evidenced by SRF decreasing from $3.10 \times 10^{13}$ m/kg to $1.59 \times 10^{12}$ m/kg [19]. High solid concentrations of sludge can lower bioleaching performance [10]. Clearly, in engineering practices, the dewaterability of ADS might be enhanced through dilution, which would still require a lengthy bioleaching process and definitely increase operation costs. It is well known that bioleaching is an aerobic process, as aeration is necessary to support the growth of the *Acidithiobacillus* species. Probably, such sudden environmental change from anaerobic to aerobic states due to aeration induced by bioleaching may cause the death of anaerobic microorganisms in ADS by releasing intracellular organic matter or large amounts of EPS to resist stress conditions [20,21]. This result might indicate that, unlike UDS, the dewaterability of ADS is not easily improved to a desired level through bioleaching, a condition that might be related to the aeration. However, to the best of our knowledge, the reasons for the ineffectiveness of enhancing the dewaterability of ADS by bioleaching remain to be discovered.

This study aimed at figuring out the reasons for the ineffectiveness of enhancing the dewaterability of ADS by bioleaching. The dewatering performances between ADS and UDS by bioleaching were systematically compared. Moreover, the influence of aeration on the dewaterability of UDS and ADS and the properties of sludge EPS were illuminated. An understanding of this study would be helpful in seeking a more efficient strategy for improving dewaterability of ADS by bioleaching.

## 2. Materials and Methods

### 2.1. Sewage Sludge Samples

The samples of UDS with three replications were randomly collected from the thickened sludge pool of the Taihu New City Wastewater Treatment Plant in Wuxi City, Jiangsu Province, China. The UDS samples were transported to the laboratory in polypropylene containers, and then mixed together and concentrated across gravity settling at 4 °C. ADS was prepared by batch anaerobic digestion. Briefly, 10 L of UDS was placed in an anaerobic digester, then anaerobically digested at 38 °C for more than 20 days. The anaerobic digester was a cylindrical box made of perspex glass with a total volume of 15 L. Before the experiment, UDS was diluted with distilled water to produce the same solid content as that of ADS. The preliminary physicochemical properties of UDS and ADS before bioleaching are presented in Table 1.

**Table 1.** Preliminary physicochemical characteristics of UDS and ADS before bioleaching.

| Items | UDS | ADS |
|---|---|---|
| pH | $6.68 \pm 0.01$ | $7.10 \pm 0.01$ |
| TS (%) | $3.07 \pm 0.02$ | $3.00 \pm 0.01$ |
| VSS (%) | $54.6 \pm 0.1$ | $48.9 \pm 0.3$ |
| Organic matter content (%) | $54.8 \pm 0.1$ | $49.3 \pm 0.2$ |
| SRF ($\times 10^{12}$ m/kg) | $6.40 \pm 0.03$ | $60.80 \pm 0.1$ |
| CST (s) | $23.6 \pm 0.1$ | $93.1 \pm 0.3$ |
| EPS (mg/g-VSS) | $4.5 \pm 0.1$ | $7.2 \pm 0.5$ |

### 2.2. Bioleaching Inoculum Preparation

Acidophilic chemoautotrophic bacterium *Acidithiobacillus ferrooxidans* LX5 (CGMCC No. 0727) obtained from China General Microbiological Culture Collection Center (CGMCC) was cultured in modified 9K medium [12]. The modified $Fe^{2+}$-free 9K medium was acidified with sulfuric acid to pH 2.5, then autoclaved at 121 °C for 15 min. An amount of 50 mL of inoculum was added into 1 L Erlenmeyer flasks containing 25 mL of modified 9K medium and 425 mL of 0.22 μm membrane-filtered $FeSO_4 \cdot 7H_2O$ (52.0 g/L) and cultured on a rotary shaker at 28 °C and 180 rpm.

The amount of 60 mL of cultures of *A. ferrooxidans* LX5 was added in 500 mL Erlenmeyer flasks containing 240 mL of UDS and 10 g/L of $FeSO_4 \cdot 7H_2O$. Then the flasks were incubated in a rotary shaker at 28 °C at 180 rpm. When the system pH was less than 2.0, the 60 mL acidified bioleached sludge was transferred to a new flask containing 240 mL of UDS and 10 g/L of $FeSO_4 \cdot 7H_2O$, as described above. After two more rounds of transfer and incubation, freshly acidified, bioleached sludge was employed as inoculum in the following experiments.

### 2.3. Bioleaching Experiments

First, 450 mL of UDS or ADS sludge was placed into a series of 1 L Erlenmeyer flasks. Then, 50 mL of bioleaching inoculum and $FeSO_4 \cdot 7H_2O$ at a dose of 10 g/L were added to each flask. Bioleaching was performed in a rotary shaker at 28 °C and 180 rpm. The loss of water in each flask due to evaporation during bioleaching was compensated by adding distilled water based on weight loss. All groups were designated in two sets: one set was used to measure sludge pH, and the other was sacrificed to measure the indexes of sludge dewaterability (SRF and/or CST) and $Fe^{2+}$. A 50 mL sludge sample was collected from flasks at sampling intervals. Seven rounds of sampling were conducted during bioleaching at 0 h, 1 h, 6 h, 12 h, 24 h, 36 h, and 48 h of the process. Unless otherwise stated, all treatments were conducted in triplicate in this study, and the data were presented as arithmetic mean ± standard deviations.

### 2.4. Aeration Experiments

An amount of 1 L of UDS or ADS sludge was placed in a series of 2 L Erlenmeyer flasks, which were shaken in a rotary shaker at 28 °C and 180 rpm for 144 h. The temperature and rotary speed were the same as those of the above bioleaching treatments. During the aeration, 90 mL of sludge was sampled at 0 h, 1 h, 12 h, 24 h, 48 h, 72 h, 96 h, and 144 h.

In order to assess whether the lysis of microbial cells in sludge occurred during the aeration of sludge, the organic matter of aerated sludge after EPS extraction was compared with that of raw sludge after EPS extraction. An amount of 50 mL of ADS or UDS was placed in a 150 mL Erlenmeyer flask, then aerated according to the above procedures for 24 h. After the layered EPS extraction, the residual was re-suspended with deionized water to its original mass. Then, 30 mL of the mixture was collected and determined for organic and inorganic matter contents. In addition, the contents of organic and inorganic matters in the unaerated ADS or UDS after EPS extraction were also measured to exclude the influence of the extraction method.

### 2.5. Analytical Methods

Slime EPS (i.e., soluble EPS, which are weakly bound cells or dissolved into the solution), loosely bound EPS (LB-EPS), and tightly bound EPS (TB-EPS) were extracted from sludge samples following a modified method recommended by previous studies [22]. Briefly, 30 mL of sludge samples was collected and centrifuged at $2500 \times g$ and 4 °C for 15 min. The supernatant was collected as slime EPS. The collected bottom sediments were washed twice with 0.05% NaCl solution and re-suspended to their original volume. The suspensions were centrifuged again at $5000 \times g$ and 4 °C for 15 min with the supernatant, and the solid phase was collected separately. The organic matters in the supernatant were the LB-EPS of sludge samples. Collected sediments were washed twice and re-suspended again with 0.05% NaCl solution to the original volumes, then treated using heating at 60 °C for 30 min. The extracted solutions were centrifuged at $15,000 \times g$ and 4 °C for 20 min. The organic matters in the supernatant were the TB-EPS. The collected slime EPS, LB-EPS, and TB-EPS solutions were separately passed through 0.45 μm polytetrafluoroethylene membranes and 3500 Da dialysis membranes to remove particulates and low-molecular-weight metabolites. Total organic carbon (TOC) in extracted EPS solutions was analyzed by a TOC analyzer (TOC-5000A, Shimadzu, Kyoto, JPN). Polysaccharides (PS) and proteins (PN) contents in the EPS solution were measured using the anthrone method and modified Lowry method with glucose and Bovine albumin as standards, respectively. Aqueous $Fe^{2+}$ concentration was quantified by the 1,10-phenanthroline method [23]. Sludge pH and organic matter were measured according to the Standard Method [23]. Specific resistance to filtration (SRF) was measured by using a Buchner funnel [24], and sludge capillary suction time (CST) was measured using a capillary suction timer (Model 304M, Triton, London, UK).

### 2.6. Statistical Analysis

The SPSS software was used to compare the measurement data and perform the correlation analysis. Measurement data were expressed as Mean $\pm$ Standard Deviation (SD). Student's *t*-test was used to test the difference between pairs of data sets. Statistical significance was considered as a *p*-value less than 0.05.

## 3. Results and Discussion

### 3.1. Dewaterability of UDS and ADS during Bioleaching

As shown in Figure 1a, the SRF of UDS was decreased by 91.6% to only $5.39 \times 10^{11}$ m/kg within 36 h of bioleaching, indicating a drastic improvement in the dewaterability of UDS. This result was consistent with previous studies that showed the bioleaching driven by *A. ferrooxidans* could improve the dewaterability of waste-activated sludge [17]. However, at the initial stage of bioleaching, the SRF of ADS only decreased by 63.2%, which was still as high as $2.23 \times 10^{13}$ m/kg. Subsequently, even at the end of bioleaching, the SRF of ADS

steadily kept going up to $7.28 \times 10^{13}$ m/kg, which was close to the initial SRF value of ADS. Such an increase rather than a decrease in the SRF of ADS indicated that bioleaching indeed was ineffective to improve the dewaterability of ADS. In the bioleaching process, the *Acidithiobacillus* species triggers the bio-oxidation of added-$Fe^{2+}$, and, consequently, the pH value of the matrix is lowered due to the production of $H^+$ through hydrolysis of resultant $Fe^{3+}$ [10]. Furthermore, the oxidation efficiency of $Fe^{2+}$ added as energy substances for *A. ferrooxidans* LX5 could be used to reflect the growth of that bioleaching bacterium. Thus, the changes in sludge pH and $Fe^{2+}$ concentration during bioleaching with ADS were determined to further explore the causes of its failure to improve dewaterability. As shown in Figure 1b, the pH of UDS declined from 6.68 to 2.98 within 24 h and remained at this level throughout the subsequent bioleaching process. Moreover, $Fe^{2+}$ concentrations in the system of UDS steadily decreased within 36 h of bioleaching (Figure 1c), showing that $Fe^{2+}$ bio-oxidation occurred. The pH of ADS decreased within the initial 2 h, probably resulting from the oxidation of $Fe^{2+}$ by the oxygen in the air [17]. Surprisingly, the pH of ADS climbed to 7.97 in the first 6 h and remained unchanged during the 48 h of the bioleaching period. A similar phenomenon was observed by Fontmorin and Sillanpää [19], where the pH of the sludge sample without the addition of ferrous sulfate increased from 7.5 to 8.4. This evolution could be explained by the absence of an effective power source for acid production typical of iron- or sulfur-oxidizing microorganisms' activity. Meanwhile, in the system of ADS, no $Fe^{2+}$ was detected during bioleaching, which might be because the added $Fe^{2+}$ was adsorbed onto sludge particles or precipitated as $Fe(OH)_2$ immediately in this alkaline environment. In addition, an alkaline environment usually inhibits the growth of the *Acidithiobacillus* species or even kills them [13]. Thus, the alkaline environment and lack of available $Fe^{2+}$ in the system of ADS against the growth of the *Acidithiobacillus* species during bioleaching consequently impede the improvement of its dewatering performance. However, the poor dewaterability of ADS during bioleaching could not be fully explained by the inhibited growth of *A. ferrooxidans* LX5. Since bioleaching is an aerobic process, the physicochemical properties of ADS would change under aeration conditions, affecting sludge dewaterability.

### 3.2. Influence of Aeration on the Dewaterability of UDS and ADS

During bioleaching, aeration is needed to provide sufficient oxygen to support the growth of *Acidithiobacillus* since they are obligate aerobes [25]. In this study, the changes of sludge SRF and CST during 144 h of aeration without the inoculation of *A. ferrooxidans* LX5 and the addition of energy substances were determined. It can be seen in Figure 2a,b that sludge SRF and CST of UDS soared to $3.62 \times 10^{13}$ m/kg and 76.1 s within 1 h of aeration, then gradually decreased during the rest of the aeration period. The decrease in SRF and CST indicated that the dewaterability of UDS could be promoted by aeration. Unexpectedly, either sludge SRF or the CST of ADS showed a steady growth trend during the whole aeration period, increasing by 232.2% and 593.4%, respectively, resulting in the poor dewaterability of ADS. Therefore, aeration seriously deteriorated the dewaterability of ADS but improved the dewaterability of UDS.

It is well documented that anaerobic digestion of sludge is dominated by anaerobes, suggesting that oxygen might be harmful to certain anaerobes in ADS, particularly the methanogens that produce the methane in the biogas or even cause stress responses or cell lysis [20]. Consequently, a great amount of hydrophilic extracellular polymers (EPS) would be excreted by these anaerobes as stress responses or cell lysis as shown later, which drastically reduced sludge dewaterability. In addition, pH is widely recognized to affect sludge dewaterability due to the change of sludge surface charges influenced by sludge pH [26]. For instance, the presence of $H^+$ tends to neutralize the negative charges of sludge particles to decrease the repulsive interactions between sludge particles [15], resulting in the enhancement of sludge dewaterability. As shown in Figure 2c, the pH of UDS increased from 6.68 to 7.10 within 1 h of aeration, then dropped smoothly to 4.96 during the remaining period of aeration. The pH of ADS increased from 7.10 to 8.40 within the first 24 h of aeration and stayed within an alkaline range (7.65–7.75) during the aeration

period. Previous studies reported that the decrease of sludge pH of UDS during aeration is beneficial for improving its dewaterability [5]. However, compared to the UDS, the increase in sludge pH of ADS led to the deterioration in dewaterability of aerated ADS, thereby creating more unfavorable conditions for the subsequent bioleaching.

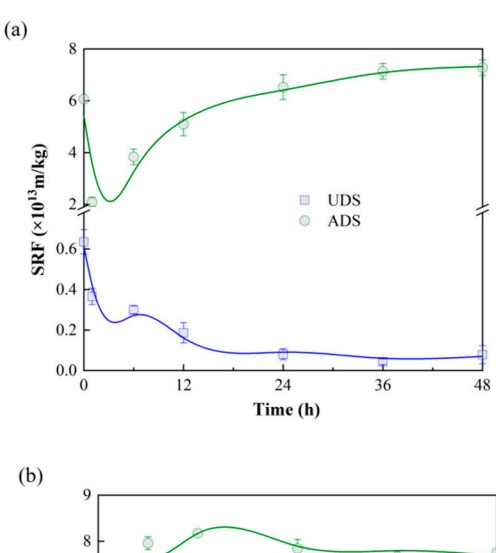

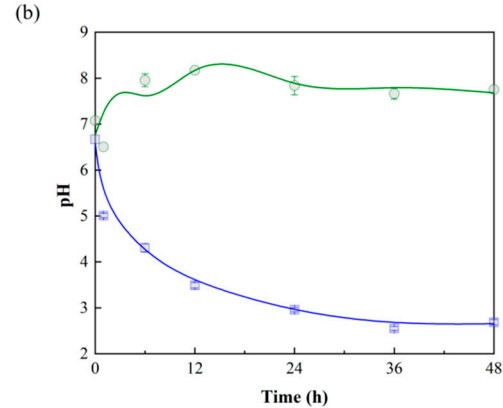

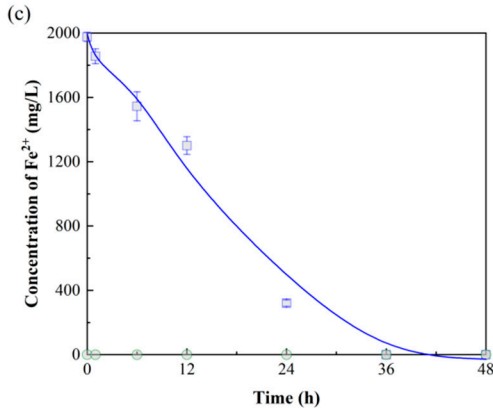

**Figure 1.** Profiles of SRF (**a**), pH (**b**), and $Fe^{2+}$ concentration (**c**) in sludge during 48 h of bioleaching with UDS and ADS.

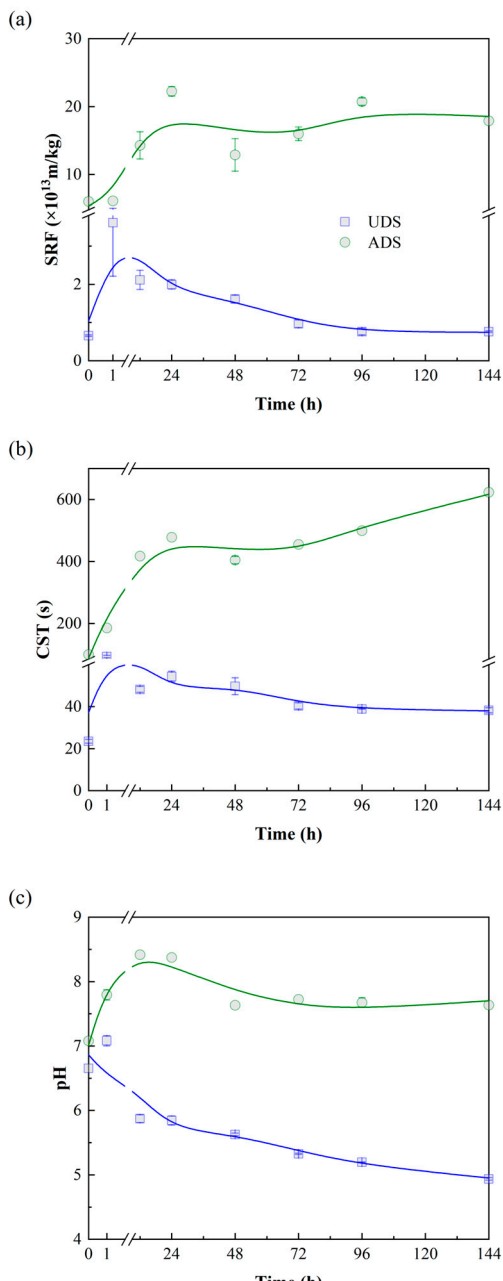

**Figure 2.** Profiles of sludge SRF (**a**), CST (**b**), and pH (**c**) during 144 h of aeration with UDS and ADS.

### 3.3. Influence of Aeration on the Properties of Sludge EPS

To figure out the influence of aeration on the properties of sludge EPS, changes of EPS concentration and composition during aeration were investigated (Figure 3). Before aeration at time 0, there was no significant difference in TB-EPS and LB-EPS contents ($p > 0.05$) between UDS and ADS, whereas their slime EPS contents were completely different ($p < 0.05$). As shown in Figure 3a, the contents of slime EPS, LB-EPS, and TB-EPS in UDS increased slightly within 1 h of aeration and decreased steadily within the remaining period of aeration, a result which was consistent with the slight deterioration of dewaterability of UDS during the same period of bioleaching (Figure 1a). Particularly, after 96 h of aeration, the contents of slime EPS, LB-EPS, and TB-EPS decreased by 77.8%, 66.3%, and 69.0%, respectively, compared to their contents after 1 h of aeration. Furthermore, the composition analysis of sludge EPS in UDS (Figure 3b,c) revealed that the decrease of sludge EPS could be ascribed to the significant decrease of PN and PS contents, which

might be biodegraded by some enzymes, such as protease, amylase [27,28] or some aerobes existing in UDS [29]. For instance, the PN content in slime EPS, LB-EPS, and TB-EPS during the aeration from 1 h to 96 h decreased from 1.58, 2.06, and 2.66 mg/g-VSS to 0.70, 1.04, and 0.86 mg/g-VSS, respectively. Meanwhile, the PS content in slime EPS, LB-EPS, and TB-EPS also decreased by 43.8–64.0%. Previous studies have found that the decrease in sludge EPS content was helpful in improving sludge dewaterability [30], in which a large number of functional groups contained in sludge EPS, such as hydroxyl, could increase the repulsion between flocs [31] and absorb plenty of bound water [32]. Therefore, the better dewatering performance of UDS could be attributed to the significant decrease in EPS during the process of aeration in which both PN and PS were degraded.

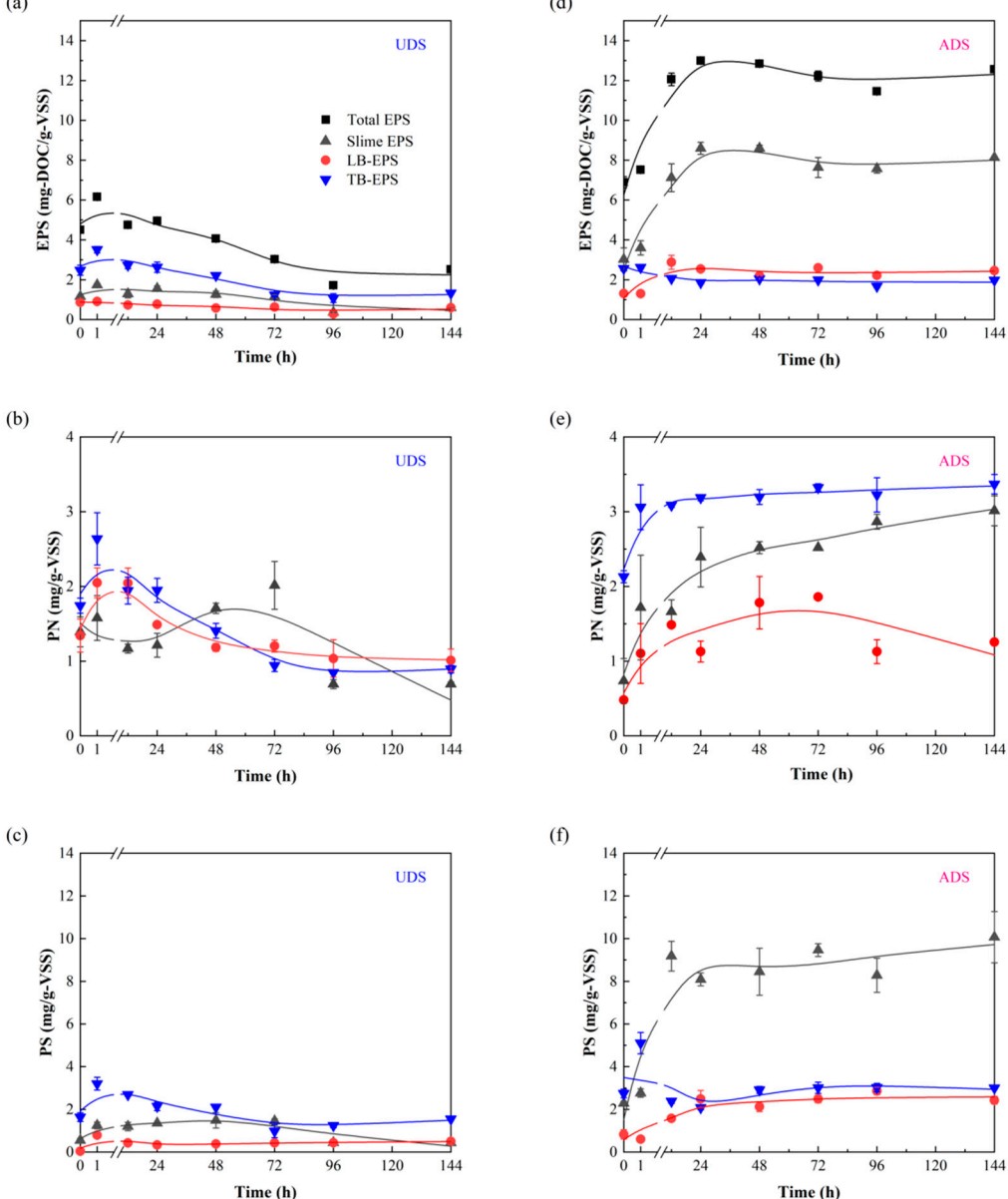

**Figure 3.** Profiles of EPS (**a**,**d**), PN (**b**,**e**), and PS (**c**,**f**) during the 144 h of aeration with UDS and ADS.

As shown in Figure 3d, the slime EPS content of ADS increased rapidly from 3.04 to 8.19 mg-DOC/g-VSS after the 144 h of aeration, while the contents of LB-EPS and TB-EPS fluctuated within the ranges of 1.32–2.47 mg-DOC/g-VSS and 1.69–2.62 mg-DOC/g-VSS, respectively. Particularly, the slime EPS, which had a significant effect on sludge dewaterability [17,33,34], accounted for about two-thirds of the total EPS of ADS. In addition,

the increase of slime EPS mainly resulted from the increases of both PN (Figure 3e) and PS (Figure 3f), which increased by 240.5% and 302.1%, respectively, within the first 72 h of aeration. Shao et al. found that CST had a significant positive correlation ($p < 0.01$) with PN and PS in the slime fraction, demonstrating that the increase of soluble organic matter could result in the deterioration of sludge dewaterability [33]. Sludge is highly complex and contains numerous types of small and surface-charged organic matter with high hydrophilicity, such as EPS [35,36]. Increased hydrophilicity generally leads to worse dewatering [37]. Therefore, it could be concluded that the dewatering performance of ADS could be affected by aeration with the significant increases of both PN and PS contents in slime EPS.

It is still unclear where large amounts of EPS originated during the aeration of ADS. According to Neyens et al. [38], EPS mainly comes from intracellular substances secreted and released by microorganisms that exist in the forms of PN, PS, and DNA. However, in the extraction procedures for sludge EPS, both secreted microbial substances and the released intracellular substances during the lysis of microbial cells could be extracted and counted as the same [22]. In this study, the increased EPS of ADS during aeration could be attributed to the secreted substances and released intracellular substances from cell lysis. In order to evaluate the contribution of cell lysis to the increase of EPS during aeration, the amount of residual organic matter in sludge pellets after EPS extraction was determined. As shown in Figure 4a, organic matter in UDS pellets after EPS extraction accounted for 56.6%, whereas it remained at 56.1% after 24 h aeration and EPS extraction, indicating that no obvious cell lysis of UDS happened after aeration. Organic matter in ADS pellets only accounted for 44.7% after 24 h aeration and EPS extraction, while after EPS extraction without 24 h aeration, it was 47.4% (Figure 4b). Obviously, the organic matter in ADS pellets decreased significantly after 24 h aeration, which could indirectly verify that significant lysis of microbial cells did occur during the aeration of ADS. Thus, the large increment of EPS in ADS during aeration was mainly due to the release of intracellular substances caused by the lysis of microbial cells.

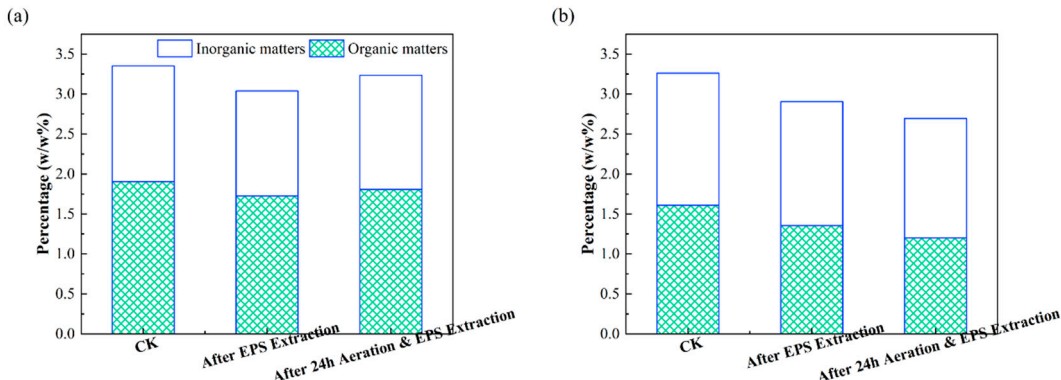

**Figure 4.** Contents of organic and inorganic matters in UDS (**a**) and ADS (**b**) after different treatments.

## 4. Conclusions

The reasons behind the ineffectiveness of bioleaching for improving ADS dewaterability were investigated in this study. The higher alkalinity and lack of available $Fe^{2+}$ in ADS created an unfavorable environment for the growth of the *Acidithiobacillus* species. In addition, the pH and slime EPS content of ADS were increased through aeration, thereby hindering the improvement of ADS dewaterability. Therefore, the ineffectiveness of sludge bioleaching for improving the dewaterability of ADS could be ascribed to both the inhibited growth of the *Acidithiobacillus* species and the deteriorated dewaterability of ADS during aeration. Nevertheless, further studies on a scale-up and continuous-flow sludge bioleaching process investigation, as well as an economic evaluation of this bioprocess, are needed for future research.



**Author Contributions:** Conceptualization, H.Z. and D.F.; Methodology, C.C.; Validation, H.Z., D.Z. and Y.Z.; Formal analysis, H.Z., D.Z. and M.Z.; Investigation, Y.Z. and B.Z.; Resources, D.Z.; Writing—original draft, H.Z. and D.Z.; Writing—review & editing, L.Z.; Visualization, D.F., J.L. (Jiansheng Li) and L.Z.; Supervision, J.L. (Jianru Liang), J.L. (Jiansheng Li) and L.Z.; Project administration, L.Z. All authors have read and agreed to the published version of the manuscript.

**Funding:** This research was funded by Science and Technology Innovation Project on Emission Peak and Carbon Neutrality of Jiangsu Province (BK20220040).

**Institutional Review Board Statement:** Not applicable.

**Informed Consent Statement:** Not applicable.

**Data Availability Statement:** Not applicable.

**Conflicts of Interest:** The authors declare that they have no known competing financial interests or personal relationships that could have appeared to influence the work reported in this paper.

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
