# Peer review of "Reasons for Ineffectiveness in Improving Dewaterability of Anaerobically Digested Sludge by Bioleaching"

_sustainability, doi:10.3390/su15064789_

Round 1
Reviewer 1 Report
Recommendation: Publish after minor revisions noted.
The authors reported the measurements of dewaterability of undigested sludge (UDS) and anaerobically digested sludge (ADS) by bioleaching under the same experimental conditions. The research idea is novel and the presented data is fundamentally important for reducing the total costs for the treatment and disposal of sludge in wastewater treatment plants. I have reviewed the manuscript carefully. Minor comments should be addressed before publication as follows.
1. As shown in Figure 1 (b), the pH of ADS decreases by (1~2 hours). Please give the reason why the pH of ADS decreases at a certain time.
2. I suggest the authors add the mechanism of the reaction especially for Fe2+.
3. Did the EPS content change during the bleaching treatment? If yes, please discuss the main factors and components in the sludge.
4. Please explain the changes in the microbial community during the bio-bleaching process
Reviewer 2 Report
The article discusses the reasons of ineffective bioleaching used to improve the dewatering of anaerobically digested sludge in comparison to undigested sludge. The article is interesting and the results clearly indicate the reasons for the low efficiency of the process. The article is well written and all data are relevant and well prepared. Congratulations!
Reviewer 3 Report
Manuscript Number: Sustainability-2171300
Full Title: Ineffective Bioleaching in Improving Anaerobically Digested Sludge Dewaterability Resulting From Restricted Growth of Acidithiobacillus Species and Deteriorated Dewaterability During Aeration.
General Comments:
The work represented here is interesting and important. However, there are a few issues that must be addressed.
Section-wise Comments:
In Title
The title is very long and highly confusing. I personally didn’t understand – what actually the article aims to describe. Please make the title interesting and informative to encourage a wide range of readers to read the article.
In Abstract
The abstract needs revision. The authors should introduce the topic first and then discuss the main findings.
In Introduction
The introduction is well written but lacks in describing -1. the novelty, and 2. the main aims and objectives of this study. I also request the authors to cite some latest references.
In the Materials and Method Section
This section is well written. However, Section 2.1 needs improvement with more details about the sampling process and sampling replicates. In Table 1, only EPS shows SE. What about the others?
In Result
The results need more statistical inputs. I think all the represented data should be analyzed through ANOVA or any suitable format. I think if the results are processed properly, then the discussion will be enriched and better understandable. In general, the authors have only concentrated their discussion by comparing the UDS and ADS only. The overall work has more dimensions, and the authors should explore that.
In Figures
The figures are a bit monotonous. The authors can try something different than only line graphs to represent different results.
Round 2
Reviewer 3 Report
I must thank the authors for these relevant revisions. In my opinion, the article is really good in its present form.